# Preclinical Study of DNA-Recognized Peptide Compound Pyrrole-Imidazole Polyamide Targeting Human TGF-β1 Promoter for Progressive Renal Diseases in the Common Marmoset

**DOI:** 10.3390/molecules24173178

**Published:** 2019-09-01

**Authors:** Masari Otsuki, Noboru Fukuda, Takashi Inoue, Takayuki Mineshige, Tomoyasu Otsuki, Shu Horikoshi, Morito Endo, Masanori Abe

**Affiliations:** 1Division of Nephrology, Hypertension and Endocrinology, Department of Internal Medicine, Nihon University School of Medicine, Tokyo 173-8610, Japan; 2Research Center Nihon University, Tokyo 101-0061, Japan; 3Marmoset Research Department, Central Institute for Experimental Animals, Kanagawa 210-0821, Japan; 4Faculty of Human Health Science, Hachinohe Gakuin University, Hachinohe, Aomori 031-8588, Japan

**Keywords:** human, TGF-β1, pyrrole-imidazole polyamide, renal disease, marmoset

## Abstract

Pyrrole-imidazole (PI) polyamides are novel gene silencers that strongly bind the promoter region of target genes in a sequence-specific manner to inhibit gene transcription. We created a PI polyamide targeting human TGF-β1 (hTGF-β1). To develop this PI polyamide targeting hTGF-β1 (Polyamide) as a practical medicine for treating progressive renal diseases, we examined the effects of Polyamide in two common marmoset models of nephropathy. We performed lead optimization of PI polyamides that targeted hTGF-β1 by inhibiting in a dose-dependent manner the expression of TGF-β1 mRNA stimulated by PMA in marmoset fibroblasts. Marmosets were housed and fed with a 0.05% NaCl and magnesium diet and treated with cyclosporine A (CsA; 37.5 mg/kg/day, eight weeks) to establish chronic nephropathy. We treated the marmosets with nephropathy with Polyamide (1 mg/kg/week, four weeks). We also established a unilateral urethral obstruction (UUO) model to examine the effects of Polyamide (1 mg/kg/week, four times) in marmosets. Histologically, the renal medulla from CsA-treated marmosets showed cast formation and interstitial fibrosis in the renal medulla. Immunohistochemistry showed strong staining of Polyamide in the renal medulla from CsA-treated marmosets. Polyamide treatment (1 mg/kg/week, four times) reduced hTGF-β1 staining and urinary protein excretion in CsA-treated marmosets. In UUO kidneys from marmosets, Polyamide reduced the glomerular injury score and tubulointerstitial injury score. Polyamide significantly suppressed hTGF-β1 and snail mRNA expression in UUO kidneys from the marmosets. Polyamide effectively improved CsA- and UUO-associated nephropathy, indicating its potential application in the prevention of renal fibrosis in progressive renal diseases.

## 1. Introduction

The academic field of chemical genomics is a chemistry-based field that involves research on genomics associated with genetic diagnostics, genome drug discovery, biochips, biomaterials, and other related topics. Thus, chemical genomics is expected to be the most important field for next-generation drug discovery. Middle molecule compounds (molecular weight 500–4000), including peptide compounds, have received attention as next-generation medicines. Middle molecule compounds have great potential as multipoint interactions with target genes and proteins in comparison to small molecule compounds. Pyrrole-imidazole (PI) polyamides, a type of middle molecule peptide, were discovered based on chemical genomics.

PI polyamides are DNA-recognized peptide compounds initially identified from antibiotics such as duocarmycin A and distamycin A. PI polyamides are aromatic rings of the amino acids *N*-methylpyrrole and *N*-methylimidazole, which recognize and bind to DNA in a sequence-specific manner [1,2]. PI polyamides have the ability to form hydrogen bonds with high affinity and specificity to double-stranded DNA (dsDNA), which is stronger in comparison to the bonds between protein and dsDNA. PI polyamides can inhibit protein interactions and DNA transcription factors [3,4]. dsDNA recognition depends on the side-by-side pairing of Pyrrole (Py) and Imidazole (Im) in the minor groove. Pairing Im opposite Py targets the G-C base pair, whereas Py-Im pairing targets the C-G base pair. Py-Py pairing targets A-T and T-A base pairs and results in degeneration [2]. Structural models of the bonding of synthetic polyamide to dsDNA are shown in Figure 1. Synthetic polyamide binds to the appropriate B-form dsDNA in the minor groove [5]. The initiation of transcription, which is essential for genetic expression, requires that transcription factors bind to the cognate DNA response elements in the gene promoter. Figure 2 shows a principle for the suppression of gene transcription. PI polyamides bind to the minor groove, and the block binding of transcription factors inhibits genetic expression. PI polyamides that are designed to bind transcription factor binding sites may potentially suppress genetic expression.

PI polyamides can fully resist biological degradation induced by nucleases. As PI polyamides can permeate cells and easily enter nuclei, they do not require vector-assisted delivery systems. Thus, PI polyamides may be more highly applicable as novel gene therapy agents than nucleic acid medicines [6]. Synthetic PI polyamides designed to target gene promoters might be effective as practical medicines to regulate gene transcription.

TGF-β comprises a large family of cytokines involved in the regulation of growth, differentiation, and morphogenesis of a wide range of cell types [7]. TGF-β1 is a multifunctional cytokine associated with cell growth, matrix formation, tissue repair, and inflammation [8]. We already reported the significant inhibition in human cells of both TGF-β1 promoter activity and the expression of TGF-β1 mRNA and protein by PI polyamides that target the human TGF-β1 (hTGF-β1) promoter [9]. We also showed that a PI polyamide targeting TGF-β1 can effectively attenuate progressive renal diseases [10,11], carotid artery stenosis after angioplasty [12], alkali burns of the cornea [13], skin hypertrophic scars [14], liver fibrosis [15], and encapsulating peritoneal sclerosis [16] in rats.

Even though long-term therapies are available, current treatments cannot cure progressive renal diseases caused by diabetic nephropathy, chronic glomerulonephritis, and glomerulosclerosis. Every year in Japan, 280,000 patients with end-stage renal failure undergo dialysis, and 37,000 patients newly receive dialysis. The medical expenses associated with dialysis represent a large problem for the medical economy. Thus, the development of a definitive treatment for chronic renal failure is a pressing issue. The PI polyamide targeting hTGF-β1 is a candidate compound for a definitive medicine in the treatment of progressive renal diseases.

To determine a lead compound among PI polyamides that targets hTGF-β1, we designed 7 PI polyamides to bind the hTGF-β1 promoter sequences [17]. We chose a lead compound based on the effects of these 7 PI polyamides on the hTGF-β1 mRNA expression in human vascular smooth muscle. We showed that a relatively low dose of the lead compound completely inhibited hypertrophic scarring on the skin of common marmosets [18].

In the present preclinical study, to evaluate PI polyamides as practical medicines in the treatment of progressive renal diseases in humans, we assessed the effects of a PI polyamide targeting hTGF-β1 on the progression of renal injury in common marmosets.

## 2. Results

### 2.1. Effects of Polyamide on Fibrosis in Cyclosporine A (CsA)-Induced Nephropathy in Marmosets

Figure 3 shows Masson trichrome staining of the renal medulla in marmosets injected with CsA with or without Polyamide. CsA (40 mg/kg) apparently enhanced Masson trichrome staining in the renal medulla in marmosets. Polyamide (1 mg/kg/week for five weeks) inhibited staining in the renal medulla, indicating that Polyamide treatment effectively improved CsA-induced renal fibrosis.

### 2.2. Effects of Polyamide on TGF-β1 Expression in CsA-Induced Nephropathy in Marmosets

Figure 4 shows immunohistochemical staining of TGF-β1 in the renal medulla and cortex from marmosets injected with CsA and treated with or without Polyamide. CsA (40 mg/kg) apparently enhanced TGF-β1 staining in the renal tubules and glomeruli in marmosets. Polyamide (1 mg/kg/week for five weeks) completely inhibited TGF-β1 staining in the renal medulla in marmosets.

### 2.3. Effects of Polyamide on Renal Degeneration in Marmoset Kidneys with Unilateral Urethral Obstruction (UUO)

Figure 5 shows HE and Masson trichrome staining of the renal medulla in the contralateral unobstructed kidney (CUK) and marmoset kidneys with UUO. Marmoset kidneys with UUO showed renal medulla degeneration consisting of tubular atrophy, interstitial fibrosis, and cast formation in contrast to the CUK. Polyamide (1 mg/kg/week for five weeks) inhibited degeneration and fibrosis of the kidney with UUO (Figure 5A). Polyamide treatment did not significantly suppress the glomerular injury score (GIS), whereas it did significantly (*p* < 0.05) suppress the tubulointerstitial injury score (TIS) in marmoset kidneys with UUO (Figure 5B,C).

### 2.4. Effects of Polyamide on the Expression of TGF-β1, α-Smooth Muscle Actin (α-SMA), and E-cadherin in Marmoset Kidneys with UUO

Figure 6 shows immunohistochemical staining of TGF-β1, α-SMA, and E-cadherin in the renal medulla from CUK and UUO kidneys in marmosets. Immunostaining of TGF-β1 and α-SMA were enhanced in the renal medulla of the UUO kidney in comparison to the CUK kidney. Polyamide (1 mg/kg/week for five weeks) completely inhibited the increased immunostaining of TGF-β1 and α-SMA in the renal medulla of the UUO kidney. In contrast, immunostaining of E-cadherin was suppressed in the renal medulla of the UUO kidney in comparison to the CUK kidney. Polyamide (1 mg/kg/week for five weeks) preserved E-cadherin immunostaining in the renal medulla of the UUO kidney.

### 2.5. Effects of Polyamide on the Expression of TGF-β1 and Snail mRNAs in Marmoset Kidneys with UUO

Figure 7 shows the expression of TGF-β1 and Snail mRNAs in the renal medulla of CUK and UUO kidneys in marmosets. The abundance of TGF-β1 (*p* < 0.01) and Snail (*p* < 0.05) mRNAs was significantly higher in the renal medulla of the UUO kidney in comparison to the renal medulla of the CUK kidney. After Polyamide treatment, the abundance of TGF-β1 and Snail mRNAs decreased significantly (*p* < 0.01) in the UUO kidney in comparison to the untreated UUO kidney.

## 3. Discussion

In this study, we created two types of nephropathy in common marmosets: CsA-induced nephropathy and UUO nephropathy. CsA has been reported to induce acute and chronic nephropathies. The acute nephropathy is reversible and induced by the contraction of afferent arterioles with an increase in intracellular calcium ion, increased endothelial damage due to the release of endothelin, and decreases in vasodilated prostaglandin E_2_ and I_2_ and nitric oxides [19]. The chronic administration of CsA mainly induces nephrotubular injury through increases in TGF-β1 and is associated with arteriole degeneration, focal glomerulosclerosis, and interstitial fibrosis due to hyperfiltration with a decreased number of nephrons [20,21]. These histopathological changes in CsA-induced nephropathy are known to be enhanced with lower blood levels of magnesium and sodium [22]. In this study, we, therefore, created CsA-induced nephropathy in common marmosets fed a low sodium and low magnesium diet. In contrast, the spontaneous onset of nephropathy, such as the IgA nephropathy named Wasting marmoset syndrome, has been reported in individual common marmosets showing variable kidney degeneration [23]. In this study, Polyamide (1 mg/kg/week for five weeks) effectively improved the CsA-induced renal fibrosis and completely inhibited TGF-β1 staining in the marmoset renal medulla.

We next investigated the effects of a PI polyamide that targeted hTGF-β1 in marmoset kidneys with UUO, which is a standard model of renal fibrosis. In the UUO kidney, histological changes and immunostaining of target molecules can be compared to CUK kidney in the same animal. In the UUO kidney, renal tubules stimulate the production of TGF-β1, which induces the epithelial-mesenchymal transition of the renal tubules and interstitial fibrosis [24,25]. In the present study, treatment with the PI polyamide that targeted hTGF-β1 significantly suppressed fibrosis of the renal medulla, with decreases in immunostaining of TGF-β1 and α-SMA, increased immunostaining of E-cadherin, and inhibited expression of TGF-β1 and snail mRNAs in the marmoset kidneys with UUO. These findings indicate that the PI polyamide that targeted hTGF-β1 effectively suppressed renal fibrosis and the epithelial-mesenchymal transition phenomenon in renal tubules in human nephropathies and that this PI polyamide could potentially be developed as a practical medicine for human glomerulonephritis.

To evaluate the characteristics of PI polyamides as practical medicines, we previously investigated their pharmacokinetics, including their binding to nuclei and chromosomes. FITC-labeled PI polyamide was strongly localized in the nuclei of renal tubules and glomeruli at 24 h after injection in rats. The PI polyamide was also localized in nuclei of mid-layer smooth muscle cells of the aorta, lung, and liver without any drug delivery system. A high-performance liquid chromatography (HPLC) analysis of FITC-labeled PI polyamide also showed clear detection of PI polyamide in the urine, kidney, and aorta, but not in the heart or brain [10]. Thus, PI polyamides themselves can be delivered to the organs and strongly bind to the nucleus, which is a potential advantage compared to nucleic acid medicines such as siRNA and antisense DNA, which require certain drug delivery systems. After the intravenous administration of FITC-labeled PI polyamides in rats, PI polyamides were still present in the renal tubule nuclei on day 14, indicating that the once-a-week administration of PI polyamides may be effective in treating progressive renal diseases [11]. We also examined PI polyamide binding in chromosomal DNA and found its binding pattern in the chromosomal DNA in HeLa cells to be obviously different from the DAPI on the chromosome, indicating sequence-specific binding of the PI polyamides to the chromosome [11]. We then analyzed the pharmacokinetics of the PI polyamide. The pharmacokinetic parameters of PI polyamides in the intravenous dose ranges are linear, as revealed by the fact that the area under the blood concentration time curve increased linearly as a function of the dose [26]. PI polyamides were mainly excreted in the urine, and low molecular weight PI polyamide was partially excreted in bile [27].

Recently several groups have developed alkylating PI polyamides as anticancer agents [28,29,30]. DNA alkylating agents can damage cellular DNA that leads to anticancer activity. As non-specific DNA alkylation damages normal cells, alkylating PI polyamides can damage cancer cells in sequence-specific alkylation for cancer genes [31]. However, the off-target effects of the alkylating PI polyamides by binding and alkylating to non-target genes have been a concern as practical medicines. However, we developed non-alkylating PI polyamides as a gene silencer to block transcription factor binding on the target gene. Transcriptional silencing by PI polyamides may also have off-target effects by non-specific binding on a non-target gene. However, transcriptional silencing PI polyamides simply suppress the transcriptionally activated gene function in a disease state, even when binding to non-target genes, indicating that the non-alkylating PI polyamides might have low side effects and high specificity.

In conclusion, in this preclinical study, we assessed the effects of a low-dose treatment with a PI polyamide targeting hTGF-β1 that was injected once a day in common marmoset models of chronic nephropathy induced by CsA and UUO. The synthetic PI polyamide that targeted hTGF-β1 effectively improved nephropathy in common marmosets. These results indicate that PI polyamide could potentially be developed as a medicine that may provide a radical cure for progressive renal diseases.

## 4. Materials and Methods

### 4.1. Design and Synthesis of PI Polyamides Targeting the hTGF-β1 Promoter

Figure 2A illustrates the results of an NCBI BLAST Two Sequence Analysis of the hTGF-β1 promoter sequence, which showed a sequence homology of 86% between humans and common marmosets. We designed Polyamide to bind bp 819–826 next to the adipocyte P2 gene, which contains a regulatory element (FSE2) binding site of the hTGF-β1 promoter sequence (Figure 2B). The binding sequence of Polyamide was completely identical to the TGF-β1 promoter sequences of human and marmoset. The structures of Polyamide and mismatch PI polyamide are shown in Figure 2C. Mismatch PI polyamide had an identical structure and molecular weight to Polyamide.

We induced substitution of Im and Py to create the PI polyamides by machine-assisted automatic synthesis of hairpin-type PI polyamides with use of a PSSM-8 continuous-flow peptide synthesizer (Shimadzu, Kyoto, Japan) at 0.1 μmol scale (200 mg of Fmoc-b-alanine-CLEAR Acid Resin, 0.50 mEq/g, Peptide Institute, Osaka, Japan). We performed automatic solid-phase synthesis by first washing with dimethylformamide (DMF), then removing the Fmoc group with 20% piperidine/DMF, washing with methanol, coupling with a monomer for 60 min in an environment containing 1-(bis(dimethylamino)methylene)-5-chloro-1*H*-benzotriazolium 3-oxide hexafluorophosphate (HCTU) and diisopropylethylamine (4 eq each), another washing with methanol, protecting with acetic anhydride/pyridine, and finally, another wash with DMF. After removal of the Fmoc group from the Fmoc-β-alanine-Wang resin, the resin was successively washed with methanol. The coupling step was performed with Fmoc-amino acid, followed by another methanol wash. We repeated these steps until all sequencing was complete. After the coupling steps were completed, the *N*-terminal amino group was protected and washed with DMF, and the reaction vessel was drained. Next, after the cleavage step (5 mL of 91% trifluoroacetic acid-3% triisopropylsilane-3% 5 dimethylsulfide-3% water/0.1 mmol resin), we used cold ethyl ether precipitation to isolate the synthetic polyamides. After another the cleavage step (5 mL of *N*,*N*-dimethylaminopropylamine/0.1 mmol resin, 50 °C overnight), the synthetic polyamides were isolated again by cold ethyl ether precipitation. The polyamides were purified by HPLC using a PU-980 HPLC pump, UV-975 HPLC UV/VIS detector (Jasco, Easton, MD), and a Chemcobound 5-ODS-H column (Chemco Scientific, Osaka, Japan).

### 4.2. Cell Culture

We maintained the marmoset fibroblasts in Dulbecco’s modified Eagle’s medium (DMEM) supplemented with 10% fetal calf serum (Invitrogen, Carlsbad, CA, USA) and 50 mg/mL streptomycin (Invitrogen). After reaching confluence (within 7–10 days after seeding 1 vial of 10^5^ cells/cm^2^), a typical hill-and-valley pattern was observed in the vascular smooth muscle cell culture. To prepare the marmoset fibroblasts, newborn skin was digested overnight with 5 mg/mL collagenase type I (Sigma, St. Louis, MO, USA). The fibroblasts were then cultured in DMEM supplemented with 10% fetal calf serum, 0.05 mg/mL gentamicin, and 0.1 mg/mL penicillin. The cells were passaged by trypsinization with 0.05% trypsin (Gibco Life Technologies, Gaithersburg, MD, USA), plated at a density of 10^5^ cells/cm^2^ in 6- or 24-well culture dishes, and cultured in a water-saturated CO_2_ incubator at 37 °C.

### 4.3. RNA Extraction and Real-Time PCR

Total RNA was extracted from the cultured cells with TRIzol reagent (Invitrogen, CA, USA), and a Takara RNA PCR Kit (AMV) Ver. 3.0 (Takara Bio, Ohtsu, Japan) was used to reverse transcribe the total RNA (1 µg) into cDNA with random 9-mers. Assay-on-Demand primers and probes (human TGF-β1: Hs00998133_m1) were obtained from Applied Biosystems Life Technologies (Tokyo, Japan), and an ABI Prism 7300 (Applied Biosystems) was used to quantify the mRNA. Each sample (each reaction, 5 µL complementary DNA, total volume, 25 µL) was run in triplicate. We determined 18S ribosomal RNA levels with TaqMan Ribosomal RNA Control Reagents (Applied Biosystems) to control sample loading. The amplification conditions were 50 °C for 2 min, 95 °C for 10 min, 60 cycles of denaturation (95 °C for 15 s), and combined annealing-extension (60 °C for 1 min). After determining the threshold cycle (Ct), we calculated the relative quantification of the marker gene mRNA expression by the comparative Ct method.

### 4.4. Ethics and Animals

This study conformed to the standards of the US National Institute of Health’s Guide for the Care and Use of Laboratory Animals (NIH Publication No. 85-23, revised 1996). The study was approved by the Nihon University IACUC committee (approval no. AP13D009) and was conducted according to the Guidelines for Conducting Animal Experiments of the Central Institute for Experimental Animals (CIEA).

The 15 male common marmosets (*Callithrix jacchus*) used in this study were obtained from CLEA Japan, Inc. (Tokyo, Japan). Experiments were performed in the CIEA. The marmosets were housed in pairs in stainless steel living cages (39655670 cm) at 45–55% humidity and 12 h of illumination per day. A platform for a bed and wood perches for locomotion were placed in each cage to enhance the marmosets’ environment, and each cage contained a puzzle feeder. A balanced diet (CMS-1M, CLEA Japan Inc., Tokyo, Japan) that included mixed L(+)-ascorbic acid (Nacalai Tesque, Tokyo, Japan), vitamins A, D3, and E (Duphasol AE3D, Kyoritsu Seiyaku Co., Ltd., Tokyo, Japan), and honey (Nihon Hachimitsu Co., Ltd., Gifu, Japan) kept the marmosets healthy and well-nourished. The marmosets ate chow moistened with hot water in the morning and dry chow in the afternoon. The marmosets also received sponge cakes, biscuits, or apple jelly when in contact with humans. The marmosets had ad libitum access to tap water from feed valves.

### 4.5. Creation of Nephropathy in Marmosets and Treatment with PI Polyamides

We created two types of nephropathy in marmosets. Male marmosets weighing 300–350 g housed and fed with a 0.05% NaCl and 0.05% magnesium diet were subcutaneously injected with 40 mg/kg/day CsA (Novartis Pharmaceuticals, Basel, Switzerland) for eight weeks to establish chronic nephropathy. Marmosets with CsA-nephropathy were intravenously injected with Polyamide (1 mg/kg/week for four weeks) from four weeks after the start of CsA injection. To create the UUO model, marmosets were pre-anesthetized with an intramuscular injection of medetomidine 0.04 mg/kg (Nippon Zenyaku Kogyo, Koriyama, Japan), midazolam 0.40 mg/kg (Astellas Pharma, Tokyo, Japan), and butorphanol 0.40 mg/kg (Meiji Seika Pharma, Tokyo, Japan). Ampicillin 15 mg/kg (Meiji Seika Pharma) was also administered, and the animals were subcutaneously hydrated with 2 mL/head of fluid (KN No. 1 injection, Otsuka Pharmaceutical, Tokyo, Japan). The animals were subsequently anesthetized by inhalation of 1.0–3.0% isoflurane (Abbott Japan, Tokyo, Japan) via a ventilation mask. In the UUO marmosets, the left ureter was ligated at two points with 6–0 silk and severed between the two ligatures. The left ureter in the sham-operated marmosets was not disturbed. The UUO marmosets were intravenously injected with Polyamide (1 mg/kg/week) dissolved in 500 μL of H_2_O through the tail vein for four weeks from one week after the operation. Control marmosets with UUO were intravenously injected with the same amount of saline through the tail vein.

### 4.6. Histopathological and Immunohistochemical Examinations

Tissue samples from the kidney were fixed in 10% neutral buffered formalin solution and then paraffin embedded. Then, 4-µm thick sections were stained with hematoxylin-eosin or Masson trichrome. Fifty glomeruli in each section were randomly selected to determine the amount of matrix in the glomeruli. The percentage of mesangial matrix in each glomerulus was estimated and scored from 0 to 4 as follows: 0 normal; 1, up to 25% involvement of the glomerulus; 2, 25–50% involvement of the glomerulus; 3, 50–75% involvement of the glomerulus; or 4, 75–100% involvement of the glomerulus. The GIS was graded as described previously [32]. Twenty areas in each renal cortex were randomly selected for quantification of the tubulointerstitial area. Basement membrane thickening, dilation, atrophy, interstitial fibrosis, interstitial inflammation, tubular necrosis, desquamation, and hydropic degeneration were graded with the TIS as follows: grade 0, none; grade 1, <10%; grade 2, 10–25%; grade 3, 26–50%; grade 4, 51–75%; and grade 5, >75% in an average of 20 fields per kidney coronal section.

Sections of 4-µm thickness were deparaffinized, dehydrated using a routine procedure, and incubated overnight at 4 °C with TGF-β1 anti-human TGF-β1 rabbit polyclonal antibody (1:1000; Yanaihara, Shizuoka, Japan). Sections for TGF-β1 staining were incubated with horseradish peroxidase-conjugated anti-biotin labeling solution (ABC Elite Kit, Vector) for 30 min at 22 °C and then washed and incubated with 3,3′-diaminobenzidine (DAB) solution. Sections of 4-µm thickness were incubated overnight at 4 °C with anti-human E-cadherin rabbit polyclonal antibody (1:5; Abcam, Tokyo, Japan) or anti-human α-SMA monoclonal mouse antibody (1:1000; DAKO, Tokyo, Japan). For E-cadherin and α-SMA staining, sections were incubated with peroxidase-labeled mouse and rabbit polyclonal goat antibody (Sigma-Aldrich, Tokyo, Japan). Counterstaining was then performed before the sections were examined under a light microscope.

### 4.7. Statistical Analysis

Values are shown as the mean ± SE. The Student *t*-test was used to analyze unpaired data, and a two-way ANOVA with the Bonferroni/Dunn procedure was performed as a post hoc test. A *p*-value of <0.05 was considered to indicate statistical significance.

## 5. Conclusions

The synthetic PI polyamide targeting hTGF-β1 effectively improved nephropathy induced by CsA and UUO in common marmosets in this preclinical study. These results showed that PI polyamide has the potential to be developed as a practical medicine for the treatment of progressive renal diseases.

## Figures and Tables

**Figure 1 molecules-24-03178-f001:**
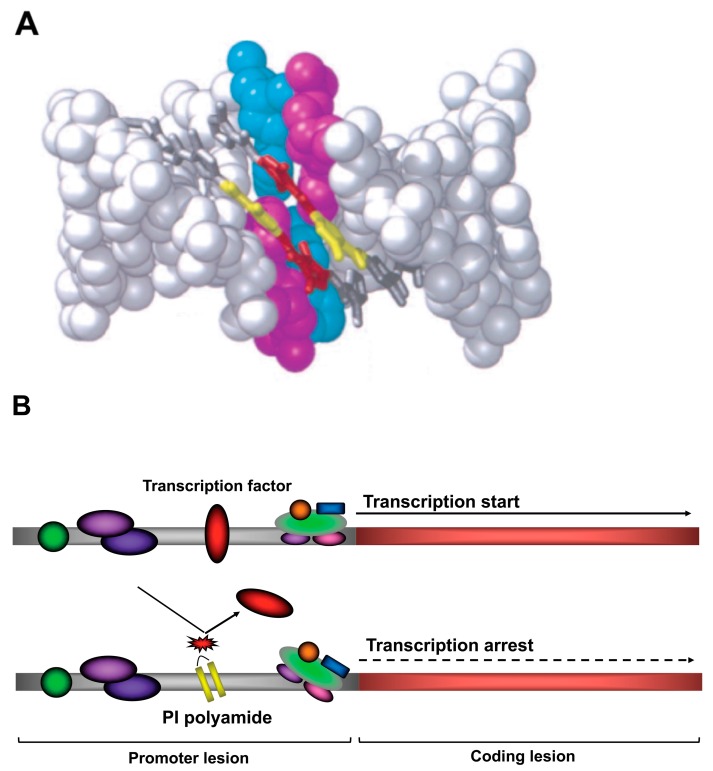
(**A**) Putative binding structures of double-stranded DNA with pyrrole-imidazole (PI) polyamide targeting human transforming growth factor-β1. Target DNA binding site is indicated by blue and pink colors. PI polyamides are indicated by yallow and red colors. (**B**) Principle of transcriptional suppression of genes. PI polyamides bind to the target promoter region of targeting genes by hydrogen binding to prevent binding of the transcription factors to transcriptionally suppress the gene expression.

**Figure 2 molecules-24-03178-f002:**
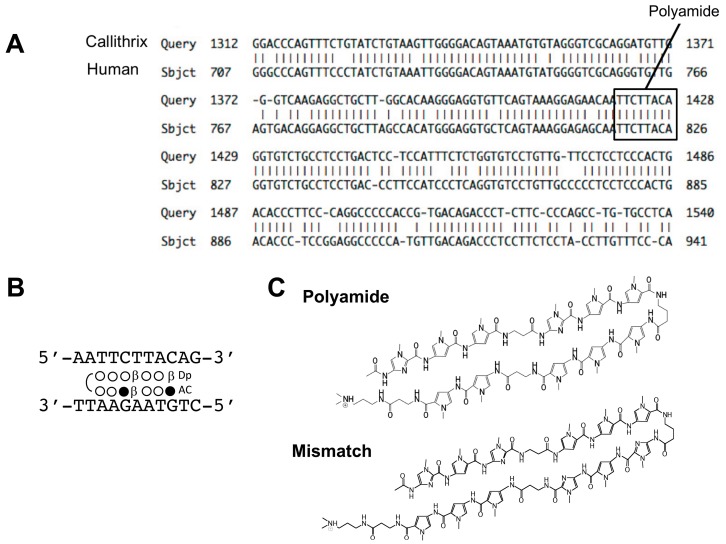
(**A**) The sequences of human and marmoset transforming growth factor (TGF)-β1 promoter analyzed by an NCBI BLAST Two-Sequence Analysis. Binding sites (box) of pyrrole-imidazole (PI) polyamide on the human and marmoset TGF-β1 promoter. (**B**,**C**) The structures and target sequences of PI polyamides targeting the human TGF-β1 promoter and the structure of the mismatch polyamide. Open circle indicates pyrrole, and closed circle indicates imidazole.

**Figure 3 molecules-24-03178-f003:**
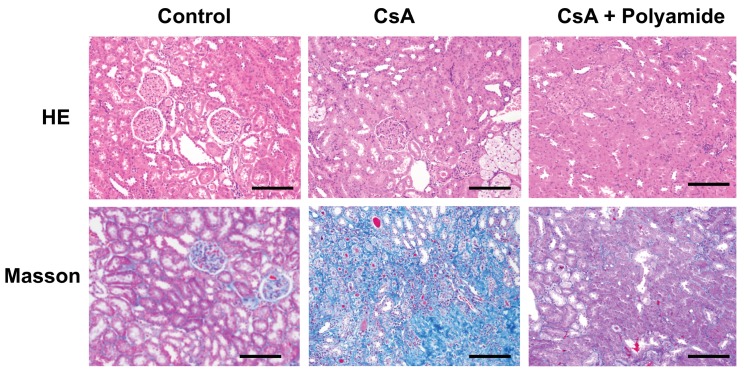
The effects of pyrrole-imidazole polyamide targeting human TGF-β1 (Polyamide) on fibrosis in marmosets with cyclosporin A (CsA)-induced nephropathy. Male marmosets were housed and fed with a 0.05% NaCl and 0.05% magnesium diet and subcutaneously injected with CsA (40 mg/kg/day) for eight weeks. Marmosets with CsA-nephropathy were intravenously injected with Polyamide (1 mg/kg/week for four weeks) from four weeks after the start of CsA injection. Sections (thickness: 4 µm) of kidney tissue samples were stained with hematoxylin-eosin (HE) and Masson trichrome (Masson).

**Figure 4 molecules-24-03178-f004:**
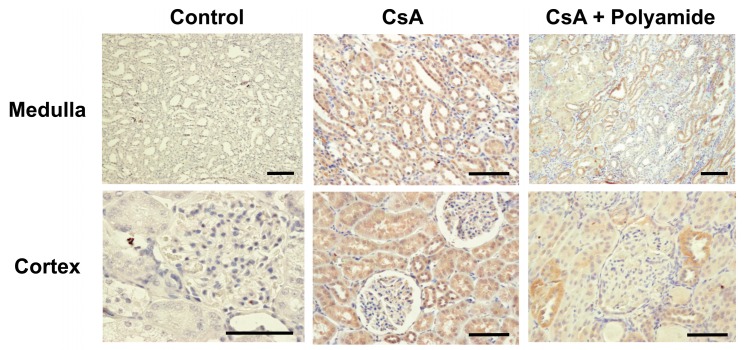
The effects of pyrrole-imidazole polyamide targeting human TGF-β1 (Polyamide) on the expression of TGF-β1 in CsA-induced nephropathy in marmosets. Male marmosets were housed and fed with a 0.05% NaCl and 0.05% magnesium diet and subcutaneously injected with CsA (40 mg/kg/day) for eight weeks. Marmosets with CsA-nephropathy were intravenously injected with Polyamide (1 mg/kg/week) for four weeks from four weeks after the start of CsA injection. Sections (thickness: 4 µm) were incubated with TGF-β1 anti-human TGF-β1 antibody and then with horseradish peroxidase-conjugated anti-biotin labeling solution.

**Figure 5 molecules-24-03178-f005:**
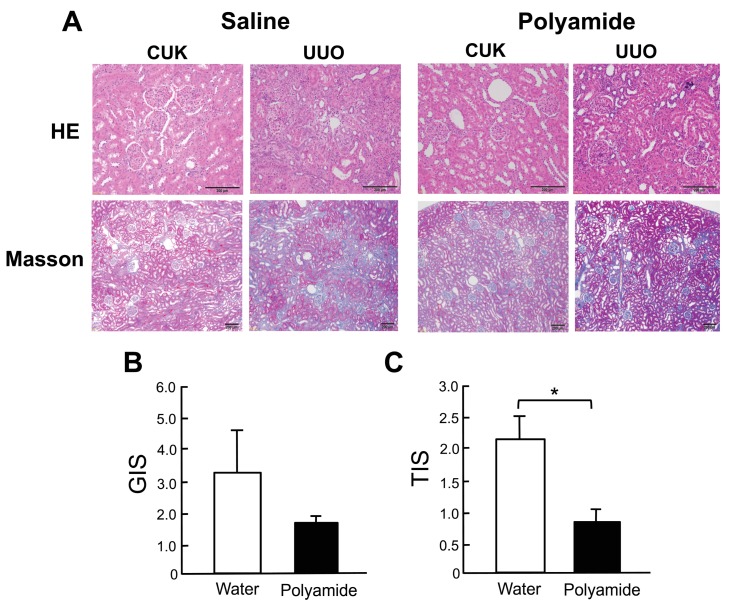
The effects of pyrrole-imidazole polyamide targeting human TGF-β1 (Polyamide) on renal degeneration in unilateral urethral obstruction (UUO) kidney in marmosets. The left ureter was ligated, whereas, in sham-operated marmosets, the left ureter was not disturbed. The marmosets with UUO were intravenously injected with Polyamide (1 mg/kg/week) through the tail vein for four weeks from one week after the operation. Control marmosets with UUO were intravenously injected with the same amount of saline through the tail vein. (**A**) Paraffin-embedded sections of the removed renal cortex were stained with hematoxylin and eosin (HE) and Masson trichrome (Masson) in the contralateral unobstructed kidney (CUK) and UUO kidney. (**B**) Glomerular injury score (GIS). (**C**) Tubulointerstitial injury score (TIS). Data indicate the mean ± SEM (*n* = 6). * *p* < 0.05 in the indicated columns. Bar = 50 μm.

**Figure 6 molecules-24-03178-f006:**
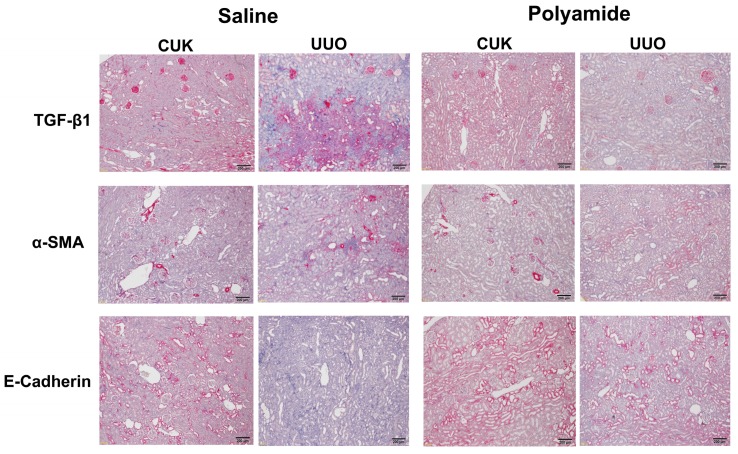
The effects of pyrrole-imidazole polyamide targeting human TGF-β1 (Polyamide) on the expression of TGF-β1, α-SMA, and E-cadherin in the kidney with unilateral urethral obstruction (UUO) in marmosets with UUO. The left ureter was ligated, whereas, in the sham-operated marmosets, the left ureter was not disturbed. Marmosets with UUO were intravenously injected with Polyamide (1 mg/kg/week) through the tail vein for four weeks from one week after the operation. CUK, contralateral unobstructed kidney. Control marmosets with UUO were intravenously injected with the same amount of saline through the tail vein.

**Figure 7 molecules-24-03178-f007:**
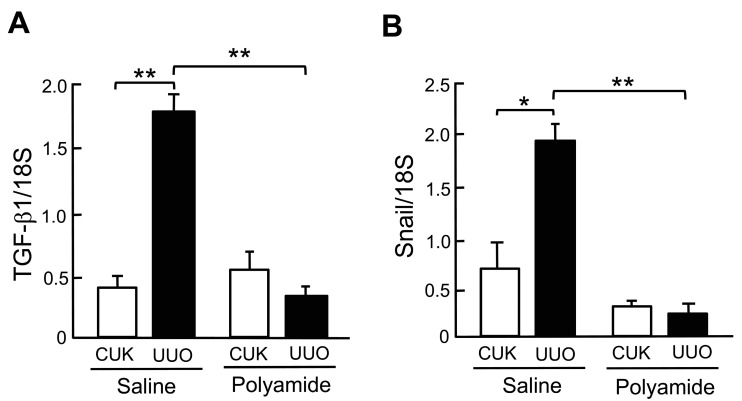
The effects of pyrrole-imidazole polyamide targeting human TGF-β1 (Polyamide) on the expression of TGF-β1 and Snail mRNAs in the kidney with unilateral urethral obstruction (UUO) in marmosets with UUO. The left ureter was ligated, whereas, in the sham-operated marmosets, the left ureter was not disturbed. Marmosets with UUO were intravenously injected with Polyamide (1 mg/kg/week for four weeks) through the tail vein from one week after the operation. Control marmosets with UUO were intravenously injected with same amount of saline through the tail vein. CUK, contralateral unobstructed kidneys. Data indicate the mean ± SEM (*n* = 6). * *p* < 0.05 and ** *p* < 0.01 in the indicated columns.

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
