# Peer review of "Preclinical Study of DNA-Recognized Peptide Compound Pyrrole-Imidazole Polyamide Targeting Human TGF-β1 Promoter for Progressive Renal Diseases in the Common Marmoset"

_molecules, 2019, doi:10.3390/molecules24173178_

Round 1

Reviewer 1 Report

In this paper, Otsuki et al present the evaluation of a pyrrole-imidazole polyamide targeting human TGF-β1 promoter in two common marmoset models of nephropathy. Although the experiments are sound and the results are worthy of publication, the entire text have to be reorganized and rewrote to make it straightforward and well-described. Other issue need to be addressed: In the Discussion section, the authors state that they investigated the pharmacokinetics (including the binding to nuclei and chromosomes), the safety study, and also the off-target effects of the PI polyamide. But they do not show any Figures and Data of these results in Results section. If these data obtained by using mice or rat model has been published in their previous work, are there any new data from the marmoset models of nephropathy?

Author Response

Reply to Reviewer 1

Thank you Reviewer’s comments to our manuscript. Editorial office indicated similarities in the text with previously published manuscripts, which we revised the similalities including title changed to “Preclinical Study of DNA-Recognized Peptide Compound Pyrrole-Imidazole Polyamide Targeting Human TGF-β1 Promoter for Progressive Renal Diseases in the Common Marmoset”

Comment: In the Discussion section, the authors state that they investigated the pharmacokinetics (including the binding to nuclei and chromosomes), the safety study, and also the off-target effects of the PI polyamide. But they do not show any Figures and Data of these results in Results section. If these data obtained by using mice or rat model has been published in their previous work, are there any new data from the marmoset models of nephropathy?

Response: Since this paper for the special issue entitled "Synthesis and Pharmacochemistry of New Pleiotropic Pyrrolyl Derivatives" in Molecules. We wrote this manuscript like a review for drug-discovery of pyrrole-imidazole polyamides, including newly ontained results of effects of PI polyamide targeting human TGF-b1 on two type of nephropathies in marmoset and our previous reports of the PI polyamide targeting TGF-b1 in rats. However, as reviewer’s comments the safety study and the off-target effects of PI polyamide targeting TGF-b1 are from our previous studies in rats, we delete the sentences of the safety study and the off-target effects of PI polyamide from Discussion section in the revised manuscript.

Reviewer 2 Report

In this manuscript, the authors describe the biological effect of a Pyrrole-Imidazole polyamide compound targeting human TGF-b1 promoter. The study is focused on the potential effect of the compound on progressive renal disease. For the study, the authors use two in vivo complementary models on marmosets.  

The rational of the study is well presented and is based on the development of PI polyamides as gene silencers to inhibit gene transcription. The study is complementary to several previous studies showing the potential of such compounds as possible future medical drug candidates (references 11 to 18) in the treatment of several diverse diseases.  

Comments:

In this study, the authors use a PI Polyamide compound which was selected according to a previous comparative study on 7 different other PI polyamide derivatives. In the present study, the term “Polyamide” to precisely name the PI polyamide used for the study is confusing. The PI polyamide compound was compound GB1101 in the PloS One 2015 study. It is suggested that the authors either keep this name GB1101 here or propose another name but not the very general generic denomination “Polyamide” for their specific compound under study.

This is also particularly confusing for the other PI polyamide used in the study which is denominated “Mismatch” when it was compound GB1106 in their 2015 manuscript with, in this former study, another fully Pyrrole polyamide compound used as mismatch comparator.

Results 2.1 and Figure 3 on the effect of the PI polyamide compound on the expression of TGF-b1 mRNA in marmoset fibroblasts in vitro were already published by the authors as Fig S2 in the PloS One 2015 study. This part should be consequently removed from the present study. Reference should be given to the previous 2015 manuscript.

Such deletion will also prevent the question on the real comparative use of such “Dismatch”, (which was compound GB1106 in the previous study) in the present study since it was previously reported by the authors as binding to the same sequences (“The binding sequences of GB1101 and G1106 were completely identical” page 6/15, PloS One 2015 – Ref 17).

The solution for in vivo injection of the PI polyamide compound should be precised.

Line 426: The phrase should be deleted since no experiment is reported in this study on hypertrophic scar creation.

Author Response

Replies to Reviewer 2

Thank you Reviewers’ comments to our manuscript. Editorial office indicated similarities in the text with previously published manuscripts, which we revised the similalities including title changed to “Preclinical Study of DNA-Recognized Peptide Compound Pyrrole-Imidazole Polyamide Targeting Human TGF-β1 Promoter for Progressive Renal Diseases in the Common Marmoset” 

Comment:In this study, the authors use a PI Polyamide compound which was selected according to a previous comparative study on 7 different other PI polyamide derivatives. In the present study, the term “Polyamide” to precisely name the PI polyamide used for the study is confusing. The PI polyamide compound was compound GB1101 in the PloS One 2015 study. It is suggested that the authors either keep this name GB1101 here or propose another name but not the very general generic denomination “Polyamide” for their specific compound under study.

Response: Since this paper for the special issue entitled "Synthesis and Pharmacochemistry of New Pleiotropic Pyrrolyl Derivatives" in Molecules. We wrote this manuscript like a review for drug-discovery of pyrrole-imidazole polyamides, including newly ontained results of effects of PI polyamide targeting human TGF-b1 on two type of nephropathies in marmoset and our previous reports of the PI polyamide targeting TGF-b1 in rats. Since this paper is like review, We use “Polyamide” as general name in this manuscript.

Comment:This is also particularly confusing for the other PI polyamide used in the study which is denominated “Mismatch” when it was compound GB1106 in their 2015 manuscript with, in this former study, another fully Pyrrole polyamide compound used as mismatch comparator. Results 2.1 and Figure 3 on the effect of the PI polyamide compound on the expression of TGF-b1 mRNA in marmoset fibroblasts in vitro were already published by the authors as Fig S2 in the PloS One 2015 study. This part should be consequently removed from the present study. Reference should be given to the previous 2015 manuscript. Such deletion will also prevent the question on the real comparative use of such “Dismatch”, (which was compound GB1106 in the previous study) in the present study since it was previously reported by the authors as binding to the same sequences (“The binding sequences of GB1101 and G1106 were completely identical” page 6/15, PloS One 2015 – Ref 17).

Response: We appreciate reviewer’s appropriate indication. Since in our designed PI polyamides targeting human TGF-b1 promoter, GB1106 did not completely affect TGF-b1 mRNA expression in marmoset-derived fibroblasts. In this study, GB1106 is setted up as mismatch PI polyamide, because mismatch polyamide should have similar structure and molecular weight of match PI polyamide. As reviewer’sindication data in Figure 3 are similar to previous data in PloS One 2015 supplement results. We delete the Figure 3 in the revised manuscript and remain statement about mismatch PI polyamide from result section.   

Comment:The solution for in vivo injection of the PI polyamide compound should be precised.

Response: We added as follows: intravenously injected with Polyamide (1 mg/kg/week) dissolved in 500 μl of H2Othrough the tail vein.

Comment: Line 426: The phrase should be deleted since no experiment is reported in this study on hypertrophic scar creation.

Response: We deleted statement No marmosets were sacrificed for the experiments involving skin hypertrophic scar creation.” from revised manuscript.”

Round 2

Reviewer 1 Report

On Page1, Line 21, The sentence "We performed lead optimization of PI polyamides that targeted hTGF-1 by inhibiting in a dose-dependent manner the expression of TGF-1 mRNA stimulated by PMA in marmoset fibroblasts." is confusing,

Reviewer 2 Report

All comments have been addressed.

The response of the authors for some of them can be considered as satisfactory and, for the other ones,  the manuscript has been modified accordingly.